# Assessment of Lymph Node Involvement with PET-CT in Advanced Epithelial Ovarian Cancer. A FRANCOGYN Group Study

**DOI:** 10.3390/jcm10040602

**Published:** 2021-02-05

**Authors:** Antoine Tardieu, Lobna Ouldamer, François Margueritte, Lauranne Rossard, Aymeline Lacorre, Nicolas Bourdel, Guillaume Lades, Camille Sallée, Jacques Monteil, Tristan Gauthier

**Affiliations:** 1Department of Gynecology and Obstetrics, CHU Limoges, 8 Avenue Dominique Larrey, CEDEX, 87042 Limoges, France; aymeline.lacorre@chu-limoges.fr (A.L.); camille.sallee@chu-limoges.fr (C.S.); Tristan.Gauthier@chu-limoges.fr (T.G.); 2Department of Gynecology and Obstetrics, Tours University Hospital, CEDEX, 37000 Tours, France; louldamer@yahoo.fr (L.O.); l.rossard@chu-tours.fr (L.R.); 3Poissy-Saint-Germain-en-Laye Hospital Center, Department of Gynecology and Obstetrics, CEDEX, 78100 Saint-Germain-en-Laye, France; fmargueritte@gmail.com; 4Department of Gynecology and Obstetrics, Clermont Ferrand University Hospital, CEDEX, 63000 Clermont Ferrand, France; nicolas.bourdel@gmail.com; 5Nuclear Medicine Department, Limoges University Hospital, 87042 Limoges, France; guillaume.lades@chu-limoges.fr (G.L.); jacques.monteil@chu-limoges.fr (J.M.)

**Keywords:** advanced epithelial ovarian cancer, positron emission tomography/computed tomography, lymphadenectomy

## Abstract

The objective of our study is to evaluate the diagnostic performance of positron emission tomography/computed tomography (PET-CT) for the assessment of lymph node involvement in advanced epithelial ovarian, fallopian tubal or peritoneal cancer (EOC). This was a retrospective, bicentric study. We included all patients over 18 years of age with a histological diagnosis of advanced EOC who had undergone PET-CT at the time of diagnosis or prior to cytoreduction surgery with pelvic or para-aortic lymphadenectomy. We included 145 patients with primary advanced EOC. The performance of PET-CT was calculated from the data of 63 patients. The sensitivity of PET-CT for preoperative lymph node evaluation was 26.7%, specificity was 90.9%, PPV was 72.7%, and NPV was 57.7%. The accuracy rate was 60.3%, and the false-negative rate was 34.9%. In the case of primary cytoreduction (*n* = 16), the sensitivity of PET-CT was 50%, specificity was 87.5%, PPV was 80%, and NPV was 63.6%. The accuracy rate was 68.8%, and the false negative rate was 25%. After neoadjuvant chemotherapy (*n* = 47), the sensitivity of PET-CT was 18.2%, specificity was 92%, PPV was 66.7%, and NPV was 56.1%. The accuracy rate was 57.5%, and the false negative rate was 38.3%. Due to its high specificity, the performance of a preoperative PET-CT scan could contribute to the de-escalation and reduction of lymphadenectomy in the surgical management of advanced EOC in a significant number of patients free of lymph node metastases.

## 1. Introduction

Epithelial ovarian, fallopian tubal or peritoneal cancer (EOC) is the eighth most common cancer in women [1]. The prognosis for this cancer is poor, with 43% overall five-year survival for all stages [2]. The treatment of EOC in France was updated in 2018 with French guidelines published by the Haute Autorité de Santé (HAS) [2].

In early-stage EOC (international federation of gynecology and obstetrics (FIGO) stage Ia-IIa), lymph node involvement ranges from 6.3% to 22% [3,4,5,6,7,8,9,10,11,12,13,14]. Pelvic and para-aortic lymphadenectomy is performed for staging purposes. Staging allows reclassification of between 8.5% and 13% of patients with evidence of lymph node involvement, and improves survival [9,12,13,15,16,17,18,19].

In advanced EOC (FIGO stage IIB-IV), node involvement ranges from 35% to 84% [15,16]. Node involvement has been considered a major prognostic factor for survival [17]. However, *the lymphadenectomy in patients with advanced ovarian neoplasms* (LION) study questioned this finding. This prospective, randomized study compared the routine performance of pelvic and lumbo-aortic lymphadenectomy with no cure in patients without suspicious pre- or intraoperative lymphadenopathy in early- and late-stage surgery [18]. This trial, despite its lack of power [19], found no impact on overall survival in patients with advanced ovarian cancer who had not undergone pelvic and lumbo-aortic lymphadenectomy. In contrast, postoperative morbidity and mortality were lower compared to those of patients who received lymphadenectomy [18]. 

Since this publication, it has been possible to avoid performing pelvic and para-aortic lymphadenectomies in the absence of clinically or radiologically suspicious lymphadenopathy [2].

The absence of lymphadenectomy would reduce the morbidity and mortality associated with the surgical management of advanced EOC in a number of patients. Reliable preoperative imaging appears to be important to avoid unnecessary lymphadenectomies in patients without lymph node involvement.

Positron emission tomography/computed tomography (PET-CT) after 18FDG injection has been shown to be appropriate for lymph node evaluation in early-stage ovarian cancer [20], locally advanced cervical cancer [21], and high-risk type 1 endometrial cancer [22]. Within our center, we have been using PET-CT for many years in the management of EOC for both the initial extension work-up and the evaluation of the response to chemotherapy instead of the thoraco-abdomino-pelvic CT scan (TAP CT). 

The objective of our study was to evaluate the diagnostic performance of PET-CT for the assessment of lymph node involvement in advanced EOC. 

## 2. Materials and Methods

### 2.1. Study Design

This was a retrospective bicentric FRANCOGYN group study. Anonymous data were extracted from the databases of the University Hospitals of Limoges and Tours from January 2008 to October 2019.

This study was approved by the Ethics Committee of the Limoges University Hospital on 21 February 2020, under number 352-2020-08. 

### 2.2. Main Objective and Main Outcome Measure

The main objective of our study was to evaluate the diagnostic performance of PET-CT for the preoperative assessment lymph node involvement in advanced stage EOC.

Our primary endpoint was the calculation of the sensitivity, specificity, positive (PPV) and negative (NPV) predictive values, and accuracy of PET-CT for preoperative lymph node evaluation in advanced stage EOC.

Preoperative PET-CT data were compared with the final pathology results of the different lymphadenectomies performed during the cytoreduction surgery.

A systematic lymphadenectomy included pelvic and para-aortic lymph node resection. The pelvic lymphadenectomy included the removal of common iliac, external, and obturator node groups. The upper limit of the para-aortic dissection was the left renal vein. The presacral, paracaval, and intercaval aortic chains were removed. The left para-aortic chain with the inframesenteric and supramesenteric groups was also removed.

### 2.3. Inclusion Criteria

We included all patients over 18 years of age with a histological diagnosis of advanced EOC who had undergone PET-CT at the time of diagnosis and/or prior to cytoreduction surgery with pelvic and/or para-aortic lymphadenectomy.

For all patients, all PET-CT scans were blindly reviewed by a nuclear medicine physician (not knowing the pathology results of the lymphadenectomies). 

### 2.4. Collected Data 

Demographic data collected from all patients included age at diagnosis, body mass index (BMI), and personal and family history of cancer.

Concerning the imaging assessment performed at the time of diagnosis and preoperatively, we recorded whether a PET-CT scan had been performed, the FIGO stage of the disease, and the presence of suspicious lymph nodes as well as their locations. We also recorded the CA 125 level at the time of diagnosis and preoperatively.

We collected data on the therapeutic strategy implemented (primary cytoreduction surgery, neoadjuvant chemotherapy (NAC), or exclusive chemotherapy).

The data collected concerning cytoreduction surgery were history of complete cytoreduction surgery (R0) or not (R1/R2) [23], the presence of palpable lymph nodes, history of digestive procedures or diaphragmatic peritonectomy, and history of pelvic or para-aortic lymphadenectomy.

Concerning the histology data, we analyzed the histological type of cancer, the number of lymph nodes collected from the different sites, and the number of metastatic nodes.

Finally, with regard to patient follow-up data, we collected the length of follow-up time, disease recurrence, and death.

The follow-up time was defined as the time interval in months between the diagnosis of the disease and the most recent assessment or death.

### 2.5. Statistical Analysis

Quantitative data were expressed as means, standard deviations, and extremes, and qualitative data as percentages. Diagnostic performance of PET-CT was evaluated according to sensitivity, specificity, positive predictive value, negative predictive value, and accuracy rate. The gold standard for this evaluation was the definitive anatomopathological examination of the para-aortic and pelvic lymph nodes.

For continuous variables, a comparison was made using a Student’s test, and for categorical variables, an exact Fisher test or a Chi2 test was used. A McNemar, Chi2, or Fisher exact test to compare the intrinsic characteristics of a test (based on whether the samples were independent) was used. Survival was estimated using the Kaplan-Meier method.

All analyses and calculations were done with STATA 15.1 IC^®^ software (StataCorp LLC, College Station, TX, USA). A *p* value < 0.05 was considered statistically significant. 

## 3. Results

### 3.1. Population

We included 145 patients with primary advanced EOC fulfilling all inclusion criteria. One hundred and seven patients (73.8%) were managed at the Limoges University Hospital and 38 patients (26.2%) at the Tours University Hospital.

The flow chart of our population is shown in Figure 1.

The mean age of our patients was 66 years (±10) [extremes: 37–88] with a mean BMI of 26.1 Kg/m2 (±6.4) [extremes: 16.8–52.8]. The mean follow-up time was 26.6 months (±21.8) [extremes: 1–107].

The average CA 125 level at diagnosis was 1635 U/mL (±3715) [extremes: 4–28112] and 78 U/mL (±143) [extremes: 4–788] in the preoperative period.

Key epidemiological and demographic data for our study population are presented in Table 1.

### 3.2. FIGO Stage

In our study, the FIGO stage of the cancer was determined using PET-CT performed at the time of diagnosis (*n* = 138).

In our population, 42.2% (*n* = 62/147) of our patients had stage IV disease based on PET-CT. 

Among the stage IV cases (*n* = 62), 37 patients had isolated supradiaphragmatic node involvement (59.7%) out of a total of 51 patients, with at least one supradiaphragmatic node involvement (82.3%). There were also 17 patients with at least one pleural invasion (27.4%), 5 patients with isolated inguinal node involvement (8.1%), 2 patients with at least one pulmonary involvement (3.2%), 2 patients with at least one hepatic involvement (3.2%), 2 patients with at least one bone involvement (3.2%), and 1 patient with isolated adrenal involvement (1.6%).

### 3.3. Treatment

Within our population, 22 patients (15.2%) underwent primary cytoreduction surgery. More than half of our patients (55.9% (*n* = 81/145)) received NAC before undergoing cytoreduction surgery, and 29% (*n* = 42/145) were treated with exclusive chemotherapy without the benefit of cytoreduction surgery. In total, nearly three-quarters of our patients (71%) could benefit from cytoreduction surgery at some point in their care (*n* = 103/145). Three-quarters of these patients (75.9%) underwent pelvic or para-aortic lymphadenectomy during their cytoreduction surgery. In 15.5% of cases (*n* = 16/103), this cytoreduction surgery was incomplete, leaving macroscopic residual disease. 

The presence of suspicious lymph nodes palpated intraoperatively was specified in only 16 operative reports (15.5%). Intraoperative palpation was consistent with pathology in 69.2% of cases. Regarding the ability to predict the presence of lymph node metastasis on palpation and intraoperative visualization, sensitivity was 100%, specificity was 42.8%, PPV was 60%, and NPV was 100%. In four cases, the absence of suspicious lymphadenopathy intraoperatively led to failure to perform lymphadenectomy. 

The main data concerning cytoreduction surgery are summarized in Table 2.

No supradiaphragmatic procedures were performed.

### 3.4. PET-CT Performance

The calculation of the diagnostic performance of PET-CT was possible in 63 patients (43.4%). In fact, only 63 patients in our study had a PET-CT scan prior to cytoreduction surgery including pelvic or para-aortic lymphadenectomy.

All PET-CT procedures for lymph node assessment are presented in the Table 3.

There was no significant difference in sensitivity and specificity between first and post-NAC surgery (*p* = 0.103 and *p* = 0.578 respectively).

It should be noted that 86 patients with advanced EOC benefited from preoperative PET-CT; 70.2% of these patients did not have any suspicious lymph node (pelvic, para-aortic or supradiaphragmatic) on PET-CT (*n* = 69). NAC (*n* = 47) resulted in the negation of lymph node involvement on preoperative PET-CT in 78.6% of cases.

## 4. Discussion

Our study highlighted the fact that regardless of node location or management sequence (NAC or primary surgery), PET-CT had high specificity for preoperative node evaluation in advanced EOC. In the case of advanced EOC management, considering the HAS 2018 recommendations [2] and following the LION study [18], specificity appeared to be the most important parameter for nodal assessment. Indeed, the fact that PET-CT had high specificity means that it was effective in confirming the absence of lymph node invasion on PET-CT in patients without lymph node metastasis. This allowed patients to avoid lymph node dissection, and thus reduced the morbidity associated with the surgical procedure. In our advanced EOC population, we could have avoided pelvic and para-aortic surgery in 80% of our patients.

In their 2012 meta-analysis of 13 PET-CT studies for EOC, Yuan et al. found that sensitivity and specificity were respectively 73% and 96% for the detection of lymph node metastases. These data were better than those published for chest, abdominal, and pelvic CT or magnetic resonance imaging (MRI) [16].

In 2014, Michielsen et al. published a prospective study comparing whole-body MRI with PET-CT and CT as part of the preoperative work-up of 22 patients with epithelial ovarian cancer [24]. For the evaluation of retroperitoneal lymph nodes, they described a sensitivity of 77%, specificity of 91%, positive predictive value of 77%, negative predictive value of 87% and MRI accuracy of 87%. The false negative rate was 3%. Diagnostic performance was exactly the same for PET-CT. For CT, the authors reported a sensitivity of 54%, specificity of 78%, positive predictive value of 50%, negative predictive value of 81%, and accuracy of 71%. The false-negative rate was 6%. Whole body MRI and PET-CT were statistically superior to the CT alone [24].

Table 4 summarizes the various data in the literature regarding nodal PET-CT evaluation for advanced EOC.

Our results showing the good specificity of PET-CT for lymph node evaluation were consistent with those of these previous studies. Though similar in design, our populations were not always comparable, which may partly explain the difference between our results. Our lower sensitivity may be explained by our higher proportion of NAC and lymph node involvement. However, in our population, we noted a decrease in sensitivity in case of interval surgery, even if it is not statistically significant, with an increase in the false negative rate. Our rate of lymph node involvement was also higher than that found in the Kitajima and Yoshida studies due to a high rate of patients with early-stage EOC [25,27].

Despite a good specificity, our false negative rate was still high, which explained the low sensitivity found in our study. This false negative rate may be explained in part by the spatial resolution of the PET-CT scan and the low tissue uptake of the radiotracer. Tumor FDG binding varies significantly according to tumor histology with low avidity in mucinous and clear cell adenocarcinomas [28,29]. The spatial resolution of PET-CT is 5 mm, which means that it cannot detect the presence of micrometastases. This explanation for the presence of false negatives has been highlighted by several authors who have studied PET-CT in various gynecological cancers [20,30]. According to Prakash et al., this limit of resolution of PET-CT could be responsible for 5% to 10% false negatives [31]. This false negative rate, which is partly related to the spatial resolution of PET-CT, ranges from 10% to 23% in cervical cancer [32,33]. In a study conducted by our team, we found a 20% false negative rate when PET-CT was used for lymph node evaluation in endometrial cancers [22]. This is a PET-CT-specific and cancer-independent limit. In our study, we did not record the size of lymph node metastases, so we cannot know what percentage of false negatives were related to the spatial resolution limit of PET-CT. The low tissue uptake of the radiotracer and the limited spatial resolution of PET-CT may explain the low sensitivity of PET-CT for nodal evaluation.

This rate of false negatives implies that if we had applied the LION study criteria [18], a certain number of our patients with histological lymph node involvement would not have benefitted from cures. In the LION study, nodal involvement was found in 55.7% of patients who underwent pelvic and lumbo-aortic lymphadenectomy. These patients, who had no suspicious adenopathy before or after surgery, had an overall survival identical to that of patients who did not receive pelvic and para-aortic lymphadenectomy. In view of these results, it seems that micrometastatic damage, not visible on imaging, does not have a significant impact on survival.

In our study, performing NAC prior to imaging did not appear to affect the efficacy of PET-CT. However, these results are to be qualified, because of the small number of patients treated in primary surgery, and should be verified by other studies. Even if a NAC could decrease the sensitivity of PET-CT, and thus lead to not performing a lymphadenectomy, some studies seem to show that the absence of lymphadenectomy after NAC does not modify overall and recurrence-free survival [34].

In our population, the percentage of patients with stage IV disease was significant. These stage IV rates were higher than the data in the literature, since, according to the 26th FIGO annual report published in 2006, the stage IV rate for EOC worldwide varies between 12% and 21% [7]. Studies that have used PET-CT in EOC staging also report a higher stage IV rate than the FIGO data and, more importantly, a higher rate using PET-CT compared to CT [24,35].

According to Kitajima et al., the main advantage of PET-CT in EOC is remote lymph node staging and detection of metastases rather than diagnosis of EOC itself [35].

## 5. Conclusions

Due to its high specificity, performing a preoperative 18FDG PET-CT scan could contribute to the de-escalation and reduction of lymphadenectomy in the surgical management of advanced EOC in a significant number of patients free of lymph node metastases, as shown in Figure 2. However, a low sensitivity related to its false negative rate may lead to the absence of lymph node surgical treatment in patients with lymph node invasion, which is contrary to the current practice of macroscopically complete surgery. It seems that the results of the LION study contradict this, since more than half of the patients who did not benefit from lymphadenectomy had a survival rate identical to that of the “systematic removal” group with suboptimal surgery.

Carried out in the context of extension assessment, PET-CT helps identify a significant proportion of supradiaphragmatic IV stages. These are correlated with a change in management strategies and with an increase in the NAC level before surgery to ensure complete cytoreduction. Based on recent literature, the role of PET-CT in the management of EOC may progress and become clearer in the coming years. The NCCN already recommends it for the diagnosis and monitoring of ovarian cancers [36].

## Figures and Tables

**Figure 1 jcm-10-00602-f001:**
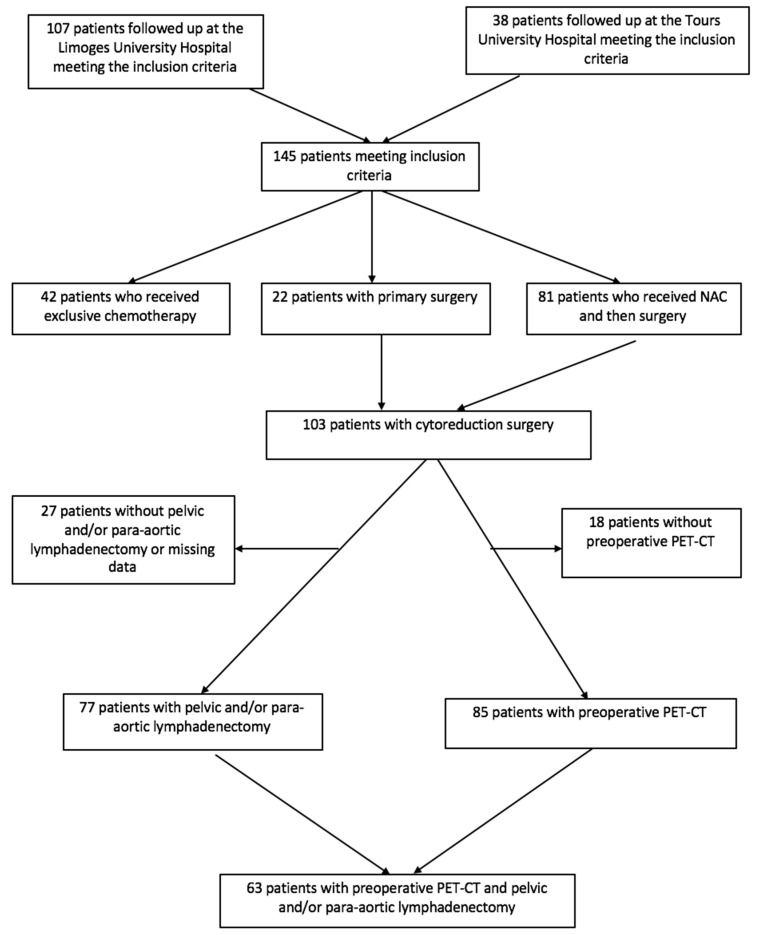
Flow chart.

**Figure 2 jcm-10-00602-f002:**
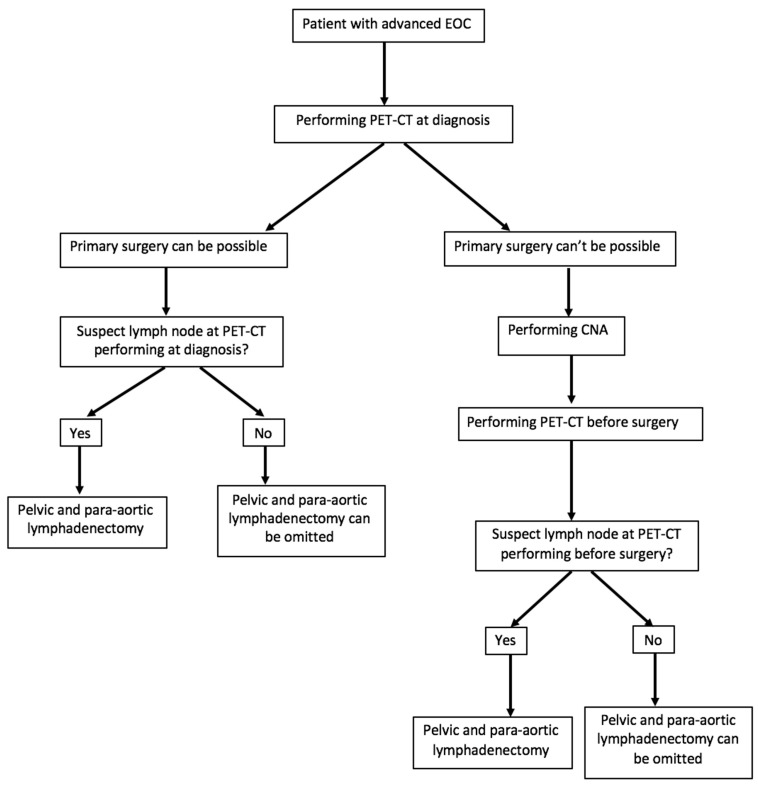
Proposal for the management of patients with an advanced epithelial ovarian, fallopian tubal, or peritoneal cancer (EOC) according to the results of positron emission tomography/computed tomography (PET-CT).

**Table 1 jcm-10-00602-t001:** Key demographic characteristics of patients (*n* = 145).

Demographic Characteristics of Patients	*n*	%
Personal history of neoplasia		
Breast	16	11
Other	7	4.8
Missing data	31	21.4
Family history of neoplasia		
Breast	27	18.6
Ovarian	6	4.1
Uterine	5	3.4
Other	25	17.2
Anatomopathology of cancer		
High-grade serous adenocarcinoma	77	53.1
Low-grade serous adenocarcinoma	11	7.6
Mucinous adenocarcinoma	5	3.4
Endometrioid adenocarcinoma	3	2.1
Clear cell adenocarcinoma	2	1.4
Carcinoma	2	1.4
Others	45	31
Survival		
Median recurrence-free survival in months	28 [1–107]	
Median overall survival in months	61 [1–107]	

**Table 2 jcm-10-00602-t002:** Key data for cytoreduction surgery (*n* = 103).

Treatment	*n*
Primary surgery (%)	22 (15.2 %)
NAC (%)	81 (55.9 %)
Mean peritoneal cancer index (standard deviation) [extreme]	8.3 (± 4.8) [0–18]
Lymphadenectomy performed (%)	
Yes	77 (74.8 %)
No	25 (24.3 %)
Missing data	1 (1 %)
Pelvic	75 (72.8 %)
Para-aortic	74 (71.8 %)
Average number of lymph nodes harvested	
(standard deviation) [extreme]	
Pelvic area	13.2 (± 7) [2–32]
Para-aortic area	21.5 (±1 5) [3–79]
Number of patients with positive lymphadenectomy (%)	
Total	36/77 (46.8 %)
Pelvic	24/75 (32 %)
Para-aortic	22/74 (29.7 %)
Missing data	1/77 (1.3 %)
Mean number of metastatic lymph nodes (standard deviation) [extreme]	
Pelvic	3.3 (± 2.7) [1–11]
Para-aortic	4.2 (± 4.2) [1–21]

**Table 3 jcm-10-00602-t003:** Performance of positron emission tomography/computed tomography (PET-CT) for lymph node assessment in advanced Epithelial ovarian, fallopian tubal, or peritoneal cancer (EOC).

	*n*	FN (%)	TN (%)	FP (%)	TP (%)	Se (%)	Sp (%)	PPV (%)	NPV (%)	Accuracy (%)
Global	63	34.9	47.6	4.8	12.7	26.7	90.9	72.7	57.7	60.3
Before primary surgery	16	25	43.8	6.3	25	50	87.5	80	63.6	68.8
After NAC	47	38.3	48.9	4.3	8.5	18.2	92	66.7	56.1	57.5
For pelvic lymph nodes	63	22.6	66.1	6.5	4.8	17.6	91.1	42.9	74.5	71
For para-aortic lymph nodes	63	21.3	65.6	1.6	11.5	35	97.6	87.5	75.5	77.1

FN: false negative; TN: true negative; FP: false positive; TP: true positive; Se: sensitivity; Sp: specificity; PPV: positive predictive value; NPV: negative predictive value.

**Table 4 jcm-10-00602-t004:** Summary of the various data in the literature regarding nodal positron emission tomography/computed tomography (PET-CT) evaluation for advanced epithelial ovarian, fallopian tubal, or peritoneal cancer (EOC).

Study	Staff	FIGO Stage Included	Lymph Nodes Sites	Se (%)	Sp (%)	VPP (%)	VPN (%)	AR (%)
Our study 2020	64	Advanced	Pelvis + para-aortic	26.7	91.2	72.7	58.5	60.9
Kitajima * [25] 2008	40	Early (50%) + Advanced (50%)	Pelvis	75	100	NR	NR	95
Nam * [26] 2010	91	Early (25.2%) + Advanced (74.8%)	Para-aortic Pelvis + para-aortic	88 83.8	94 92.6	NR 81.6	NR 93.6	93 NR
Yoshida * [27] 2004 Michielsen et al. [24] 2014	15 22	Early (33.3%) + Advanced (66.7%)	Pelvis Para-aortic Pelvis + para-aortic	100 100 77	100 100 91	100 100 77	100 100 91	100 100 87
		Advanced						

Se: sensitivity; Sp: specificity; PPV: positive predictive value; NPV: negative predictive value; AR: accuracy rate; *: Study included in the meta-analysis of Yuan et al.; NA: Not applicable; NR: Not reported.

## Data Availability

Not applicable.

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
