# Peer review of "Assessment of Lymph Node Involvement with PET-CT in Advanced Epithelial Ovarian Cancer. A FRANCOGYN Group Study"

_jcm, 2021, doi:10.3390/jcm10040602_

Round 1

Reviewer 1 Report

I have read this article with interest. The idea is good and the enrolled population is decent.

I consider this retrospective study to be a good basis for prospective studies with a larger and more homogenous population.

This study suitable for publication, but after the following major revisions:

-The use of PET CT should also be compared statistically with the current gold-standard method of intraoperative assessment of the lymph node status (manual palpation), both for upfront cytoreduction and for IDS. (Page 5, line 152; page 7, line 176). From the results obtained, discuss the possibility of improving outcomes with the combination of the two methods.

- Please discuss the reduction of sensitivity in NACTs also in light of the results of the LIANA study (doi: 10.1245 / s10434-012-2400-9. Epub 2012 May 30. PMID: 22644507.)

- Please discuss the possibility of a preoperative diagnostic flow-chart to operatively manage this patients. 

Reviewer 2 Report

The paper by Tardieu et al, “ Assessment of lymph node involvement with PET-CT in advanced ovarian cancer”  aims, in a retrospective manner, to evaluate the diagnostic performance of PET-CT for the assessment of lymph node metastasis in patients with advanced EOC. As the most important performance indicator for PET-CT, they have calculated the sensitivity, spesifity, PPV and NPV of metastatic lymph node detection with PET-CT.

The biggest problem for the reviewer is, that much  of this paper focuses on information  not relevant to the study question, and at the same time, important data regarding the study question is not reported as data, but only as results (=calculated indicators). Therefore, my suggestion is to rewrite the paper to correctly address the aim of the paper, and also give the data on basis of which the indicators are calculated.

I have spesified my thoughts more detailed below. In  addition, I have pointed out some minor remarks.

In simplicity, this kind of analysis should be done as follows:

  • the patient population should be patients, who have had a preoperative PET-CT (= the diagnostic method under evaluation) and who then have had lymphadenectomy = removal of the lymph nodes for pathological verification. In this study, there are 63such patients (line 165), and these patients constitute the study population. It is very questionable, if it is wise to combine chemonaiive and chemotherapy treated patients.

  • Only in this way, one can assess which nodes have been true positives = PET positive+ cancer, true negatives (silent nodes and no cancer pathologically), false positives (nodes turning positive in  PET-CT but being histologically normal) and false negatives (= nodes silent in PET-CT but have cancer). These numbers can then be used to calculate the performance indicators. Lines 166-172 describe the indicators  - but in order for the reader to be certain that these figures are calculated correctly, the actual numbers of PET-CT pos and neg and pathological pos and neg nodes should be given.  Actually, one should be able to assess the node individually as the true positivite or negativity of the imagined lymph node needs the histopathological verification of the finding.

Lines 70-71 …. For the preoperative assessment of lymph node involvement…

Line 89 : “ the presence and localisation of suspicious lymph nodes”. The authors tell that they have recorded this, and this is most important for the analysis as it describes the performance of the diagnostic test in question, PET-CT, but this data is not at all described for the reader in the results.

The parameters that are not relevant for the study question, such as follow-up time, age, results of cytoreduction could be removed as that is redundant information. On the other hand, one should carefully explain the extent of lymph node dissection.

Line 106: “ Diagnostic performance of PET-CT and MRI… “ MRI is not at all relevant here

Lines 156-157: the sensitivity of palpation and preoperative visualisation is reported to be 100 %. This would mean, that all pathological lymph nodes were also palpable? Sensitivity 100 % would mean that.  Was that really so? Was there selection bias as for how lymph nodes were actually chosen to be removed?

Line 178-181: I do not exactly understand the meaning. Do you mean to say, that in 69 patients, no positive lymph nodes were detected, and results of full lymphadenectomy confirmed this node negativity? From previous lines I understood, that only 63 patients hade PET-CT and surgery

Lines 180-181: Do you mean that in 47 patients, PET-CT was done prior to chemo and then after chemo but before surgery, and originally pet-positive nodes turned into pet-negative in 37 patients, and theser lymph nodes were also pathologically cancer free ?

Round 2

Reviewer 1 Report

Authors satisfied my proposed major revisions 

Reviewer 2 Report

Thank you for letting me to give my comments for this revised version.

Unfortunately, the most important comments that I made have not really been addressed.

From the paper as it is now, it is impossible to understand whether the performance indicators have been calculated accurately. The paper does not give number of removed nodes, neither number of metastatic nodes. In the Table, the metastasis/not metastasis figures seem to relate to patients, and are not given as numbers by lymph nodes.

I apologize if my review was not clear enough in meaning. it might be useful to look at some similar papers, looking at diagnostic performance and accuracy of PET-CT and see how the data usually is presented. Minimally, the numbers of removed lymph nodes and the numbers of metastatic lymph nodes are presented.

Minor comment:

- it is misleading to say in the abstract that youn included 145 patients and then give the performance indicators, when in fact there were 63 patients for this evaluation.

Lines 70-71:

Currently, the text reads " .. for the assessment of preoperative lymph node involvement" where preferably you could say " for the preoperative assessment of lymph node involvement".
